# Compounding, Rheology and Numerical Simulation of Highly Filled Graphite Compounds for Potential Fuel Cell Applications

**DOI:** 10.3390/polym15122589

**Published:** 2023-06-06

**Authors:** Alptekin Celik, Fabian Willems, Mustafa Tüzün, Svetlana Marinova, Johannes Heyn, Markus Fiedler, Christian Bonten

**Affiliations:** 1Institut für Kunststofftechnik, University of Stuttgart, 70569 Stuttgart, Germany; 2Coperion GmbH, 70469 Stuttgart, Germany; svetlana.marinova@coperion.com (S.M.); johannes.heyn@coperion.com (J.H.); markus.fiedler@coperion.com (M.F.)

**Keywords:** fuel cell, bipolar plates, compounding, twin-screw extrusion, graphite compound, CFD, simulation, particle tracking, mixing quality, wall slip, rheometry

## Abstract

Highly filled plastics may offer a suitable solution within the production process for bipolar plates. However, the compounding of conductive additives and the homogeneous mixing of the plastic melt, as well as the accurate prediction of the material behavior, pose a major challenge for polymer engineers. To support the engineering design process of compounding by twin-screw extruders, this present study offers a method to evaluate the achievable mixing quality based on numerical flow simulations. For this purpose, graphite compounds with a filling content of up to 87 wt.-% were successfully produced and characterized rheologically. Based on a particle tracking method, improved element configurations were found for twin-screw compounding. Furthermore, a method to characterize the wall slip ratios of the compounded material system with different filler content is presented, since highly filled material systems often tend to wall slip during processing, which could have a very large influence on accurate prediction. Numerical simulations of the high capillary rheometer were conducted to predict the pressure loss in the capillary. The simulation results show a good agreement and were experimentally validated. In contrast to the expectation, higher filler grades showed only a lower wall slip than compounds with a low graphite content. Despite occurring wall slip effects, the developed flow simulation for the design of slit dies can provide a good prediction for both low and high filling ratios of the graphite compounds.

## 1. Introduction

The fuel cell is an important part of the ongoing development of the automotive industry and energy sector. It has become a central subject of interdisciplinary research. Similar to a battery, a fuel cell consists of two electrodes, an anode and a cathode, which are called bipolar plates [1]. Metallic materials have the disadvantages of low corrosion resistance and low chemical resistance when producing bipolar plates. Moreover, oxides can grow on the surface due to contact transfer resistance, resulting in poor current or voltage characteristics [2].

Composite materials based on a thermoplastic polymer matrix and a very high graphite filling content (>80 wt.-%) are gaining increasing importance, especially in mobile fuel cells [3]. These composites are comparatively cheap to produce, have extremely high chemical and corrosive resistance, and are excellent in lightweight design [4]. Highly filled polymers are therefore extremely attractive for automotive applications. They combine properties such as lightweight electrical and thermal conductivity and gas non-permeability with markedly higher mechanical capacity than pure graphite plates can achieve. Moreover, they withstand the effects of moisture and acid media at typical operating temperatures of such fuel cells over longer periods of time than metal alloys do.

However, the processing of such highly filled composite materials still poses an enormous challenge. Processing by injection compression molding [5] or, more recently, by film and sheet extrusion, requires an extensive rheological characterization in the first step. Without the knowledge of the flow behavior of the highly filled polymer, molding tools, extrusion dies, and entire processing steps can only be insufficiently designed. The design process is only iteratively adjusted to the requirements by means of trial and error, with a high expenditure of time and resources. Thus, rheological characterization poses a very important and at the same time very challenging task to predict the flow behavior in polymer processing.

In polymer processing applications, many flow processes are based on the principle of wall adhesion, or this is assumed to be the case. Thus, the flow velocity of the molten polymer assumes the value zero at the wall of the flow channel. This assumption constitutes an important boundary condition for the computer-aided engineering design of polymer processing tools.

However, there are materials, such as highly filled composites, where the effect of wall slip is noticeable. In general, wall slip describes the phenomenon in which the melt flow loses its adhesion to the wall, so that shear flow and block flow occur simultaneously and overlap. If additives are added to the polymers to change their properties, e.g., as in the case of bipolar plates to increase electrical conductivity, the assumption of wall adhesion may lose its validity. This rheological phenomenon is difficult to analyze and makes the computer-aided engineering of polymer processing tools rather complicated, since the choice of boundary conditions significantly affects the obtained results.

Some important studies in recent years regarding the wall slip of polymer melts have been conducted. Even though the assumption of wall adhesion is valid for most unfilled plastics, there are some exceptions such as PVC [6,7,8,9], PE-HD [10,11,12,13], PE-LD [14], and also PLA [15]. Furthermore, wall slip effects were observed in rubber compounds. Numerous publications [16,17,18,19,20] consistently show that rubber compounds have a strong tendency to show wall slip behavior. However, different observations are made with respect to the wall slip ratios and the resulting wall slip velocities. Wall slip behavior has also been observed and investigated in the field of extrusion of ceramic compounds [21,22].

Independently from the material, two different models have been established. On the one hand, the Coulomb wall slip model is widely used, which assumes a Coulomb friction behavior between the wall and the polymer melt [11]. Here, the wall slip velocity is assumed to be a function of the friction coefficient. On the other hand, the so-called slip film model assumes that a boundary layer is actually formed between the wall and the polymer melt. The core layer is therefore more viscous than the low-viscosity surface layer and slides on the thin surface layer due to a weak adhesion.

In general, the wall slip ratios and velocities that occur during processing are difficult to quantify. Often, the relevant properties are investigated as a function of the critical wall shear stress. According to the Coulomb wall slip model, wall slip should occur above the critical wall shear stress. However, the literature on rubber compounds shows partly contradictory results. For example, Schramm [23] and Eggers [24] could not identify a critical wall shear stress, whereas Brockhaus et al. [16], Klie et al. [20], as well as Jepsen et al. [17] and Wiegreffe et al. [18] observed wall slip above the critical wall shear stress. In contrast, Geiger [25] observed that wall slip only occurs below the critical wall shear stress. His observation is based on the fact that low normal stress differences are present below the critical wall shear stress, which leads to the fact that the yield point of the cross flow towards the wall is not reached. For this reason, the surface roughness of the wall cannot be filled by the rubber compound. Above the critical wall shear stress, this cross flow will occur, which ultimately leads to wall adhesion. It is assumed that wall adhesion is influenced by the wall surface, which was investigated by Ramamurthy [26]. Recently Hatzikiriakos [27] published a good overview of existing simulation approaches and extended existing research. Despite the large number of studies, a reliable quantification of wall slip ratios and velocities remains quite difficult. There are different ways to experimentally characterize wall slip behavior. In addition to the determination of convective heat flow [12], laser Doppler anemometry [28] is also used. However, the latter is very time-consuming and can only be used for materials that are transparent in the melt phase. 

Most commonly, the quantification of wall slip is determined by rheometry. For this purpose, rotational rheometers, as well as high-pressure capillary rheometers in combination with round capillaries or slit dies, are used. The evaluation of wall slip velocity according to Mooney [29] or Mooney–Geiger [25] is based on the rheological flow curves of the material. Geiger divides the total volume flow into a shear flow and a slip flow (see Equation (1)), as well as Mooney. However, the wall slip velocity is calculated not only as a function of the wall shear stress, but also as a function of the slit height.
(1)$${\dot{V}}_{total}={\dot{V}}_{no-slip}+{\dot{V}}_{slip}=b*h*v_{s}\left(h,\;\tau_{w}\right)+b*h^{2}*A\left(\tau_{w}\right)$$

In Equation (1), ${\dot{V}}_{total}$ is the total volume rate, which consists pro rata of a slip and a no-slip volume rate. b is the width and h the height of the slit die. $v_{s}$ is the wall slip velocity. $A\left(\tau_{w}\right)$ represents the reduced yield fraction. This term is an integral function [25] describing the shear flow component and depends only on the wall shear stress. Equation (1) can be transformed as follows:(2)$$\frac{{\dot{V}}_{total}}{b*h^{2}}=F\left(h,\tau_{w}\right)*A\left(\tau_{w}\right)$$

$F\left(h,\tau_{w}\right)$ is a factor and stands for $e^{\frac{a\left(\tau_{w}\right)}{h}}$. Geiger [25] finally obtains the following function for the wall slip velocity:(3)$$v_{s}\left(h,\;\tau_{w}\right)=h*\;\left(e^{\frac{a\left(\tau_{w}\right)}{h}}-1\right)*A\left(\tau_{w}\right)$$

If a is 0, then no wall slip occurs. A disadvantage of both evaluation methods, however, is that negative slip velocities can result.

Even though the wall slip behavior for some material systems such as rubber and ceramics has already been extensively investigated, the influence of highly filled polymers with mass fractions above 80 wt.-%, such as graphite, remains unknown or insufficiently investigated. Although the flow behavior of such highly filled systems was investigated in [30], the wall slip behavior itself has not been studied. This significantly complicates the manufacturing process of bipolar plates, since the processing procedures can hardly be predicted reliably and designed accurately.

However, before the material can be rheologically characterized and processed to a sheet, a homogeneous compound, consisting of a polymer and a specific graphite filling content must be produced in the first step. The equipment which fulfills this requirement is the co-rotating twin-screw extruder. Possessing the typical modular screw design, including several conveying elements, kneading disks, and mixing elements, the extruder constantly assures the high final quality of the product. 

The aim of the powder mixing process is to give the best homogenization of all components. Nevertheless, difficulties can occur during the process, especially for new materials such as highly filled compounds. In this case, numerical simulation of the process can be very helpful to save costs and time. The development of new numerical models and techniques are essential and will strongly raise efficiency. The fluid mechanical characterization of the mixing capability of different mixing elements is one fundamental issue for efficiently designing the process and selecting an appropriate mixing element for new complex materials and higher product qualities. This can be achieved through the analysis of the fluid flow in the twin-screw extruder.

In general, the dispersive and the distributive mixing mechanism can be distinguished. The first one deals with the reduction in the minor component (fluid droplets or solid agglomeration) in the matrix or the major component, and the second one with their spatial distribution. The focus here is on the spatial distribution of the minor phase, namely the distributive mixing.

The literature shows that many methods have been used to characterize and evaluate different mixing processes, such as the extended mapping method [31], particle tracking methods, and multiphase models [32]. Researchers have used various statistical tools to characterize distributive mixing. Among them are pairwise correlation functions, the coefficient of variation or standard deviation, length stretch, Lyapunov exponent, Renyi entropies, and recently, enhancement of internal interfaces [33,34,35,36,37]. However, the most used and well-established technique remains the particle tracking method. The evaluation of the scale of segregation among bin counts [38] is one of the most used indices for the evaluation of the mixing due to its simple implementation and good results.

Simulating solid mixing processes, and particularly powder mixing, as well as evaluating the degree of mixing, is challenging due to its discretized characteristics. However, the particle tracking method was applied for the Lagrangian particle mixing concept [39], and also for the Eulerian–Lagrangian dispersive mixing process [40,41].

One of the objectives of this study is to investigate the rheological behavior of highly filled graphite compounds as well as wall slip behavior based on the theory of Geiger [25]. The novelty of this work is given by the presented study of the wall slip behavior of graphite compounds with a mass fraction of up to 87 wt.-%. In order to investigate the wall slip effects, the viscosity curves as a function of graphite content were determined by two geometrically similar slit dies on a high-pressure capillary rheometer. Subsequently, the wall slip behavior could be evaluated for the different graphite contents of the compounds. In addition, the entire experimental setup of the high-pressure capillary rheometer was transferred to a numerical simulation model in order to reliably predict the pressure loss through a slit die.

Furthermore, a combined numerical method for the evaluation of the distributive mixing process in co-rotating twin-screw extruders for highly filled polymers is proposed. First, mixing elements and kneading blocks were characterized in respect to their mixing efficiency via the particle tracking method and the statistical evaluation of the scale of segregation. Secondly, the obtained results for two element combinations were confirmed in a complex multiphase calculation, simulating the real mixing process between premixed polymer and powder in the mixing zone of the twin-screw extruders. However, the focus was based on the individual element characterization of the distributive mixing efficiency, the multiphase calculation, and the developed numerical model were mainly used for the confirmation of obtained results achieved by the particle tracking method. 

## 2. Approach and Experimental Set-Up

Figure 1 shows the steps that were necessary and carried out in this work for the production and rheological characterization of the wall slip behavior of the highly filled graphite compound. After production of the compound by means of a twin-screw extruder and rheological testing on a high-pressure capillary rheometer, the data obtained were used as input parameters in the flow simulation for the optimization of the mixing process in the twin-screw extruder. The rheological data were also used to validate a simulation model of the high-pressure capillary rheometer (HCR) together with the determined flow approach. The validation criterion was the measured pressure drop in the capillary, which was compared with the pressure drop determined from the flow simulation. The validation of the simulation model and the flow approach used served the overall objective of providing a reliable database for the future simulative design of an extrusion die for sheet extrusion of demonstrator components. The rheological analysis was performed by evaluating, for the first time, the wall slip behavior of highly filled graphite compounds according to the Geiger–Mooney correction, which is well-described in [25]. 

### 2.1. Experimental Compounding Process

Bipolar plates made of highly filled polymer compounds require a very high amount of electrically conductive fillers such as graphene, carbon black, carbon nanotubes, and so on. The incorporation of such high amounts of filler with widely varying bulk densities, especially very low bulk densities, is a major processing challenge. Therefore, special equipment is needed to feed the fillers into the extruder without reaching feeding limits. Furthermore, the material behavior of highly filled compounds can hardly be investigated by standard measurements. The segregation of different filler types with different bulk densities must be avoided during compounding. Therefore, no premixed filler combinations should be used, but each filler must be fed individually and gravimetrically into the process. At bulk densities below 0.3 g/cm^3^, large amounts of air are taken into the process, which must be reduced as much as possible. Air intakes reduce the conveying capacity, the throughput, the mixing quality of the twin-screw extruder, and finally the compound quality. Therefore, it is imperative that the air included in the fillers is minimized as far as possible before they are added to the twin-screw extruder. 

For this purpose, the fillers are fed to the twin-screw extruder via a side feeder that has a vacuum degassing by the so-called feed enhancement technology (FET). This ensures that there are more fillers, but less air is added to the process. Without this degassing process at the side feeders, the targeted fill levels could not be achieved.

Furthermore, adding all components at the twin-screw extruder inlet is not practical, because, on the one hand, feed limits are reached and, on the other hand, the polymer cannot be completely molten under these conditions. Thus, the amount of filler added must be split to different points along the process section.

For this investigation, a twin-screw extruder type ZSK26Mc18 from Coperion GmbH, Stuttgart, Germany, was used to produce the compounds. The process parameters for compounding are listed in Table 1. 

The screw has a feeding zone, a melting zone, an incorporation zone, a homogenization zone, a degassing zone, and a pressure build-up zone, as shown in the schematic diagram in Figure 2.

The polymer is added via the main feeder in the first barrel. Part of the filler is added via a side feeder with vacuum degassing before the melting zone. The remaining filler is fed to the extruder after the melting zone, also via a side feeder with vacuum degassing. To avoid segregation of the filler, no premixes are used. The different filler types and portions are fed to the side feeders by separate gravimetric feeders. After the second feeding point, all components are dispersed and homogenized in the twin-screw extruder to produce a homogenous compound. Between the two side feeders, excess air is removed from the process by atmospheric degassing. Another vacuum degassing system is also installed upstream of the discharge unit. The discharge of the highly filled compounds takes place via a hot cutting device onto a vibrating chute. The cooling of the pellets takes place in the ambient air but can also be actively accelerated by an air stream or a water-cooled vibration chute. The pellet size is defined for a given throughput by means of the blade speed and number of blades of the granulator.

With increasing filler content, the fines content in the granules typically increases. Through various equipment and process optimizations, the fines content could be reduced to a minimum. Due to the good dispersive and distributive incorporation of all filler types into the polymer matrix, the granules are very form-stable even at very high-filler contents of up to 90 wt.-%.

### 2.2. Numerical Simulation of Twin Srew Mixing Process

The particle tracking method is based on tracking of massless particles, initially located in the fluid domain. These particles simply follow the fluid flow and do not interact with each other nor with the fluid. The statistical evaluation of the distribution of these particles at the end of the considered element is its mixing characteristic. This method is used for general characterization of the distributive mixing efficiency considering one phase flow, using the final mixing viscosity. The simulation model comprises one or a combination of more elements. Only the fluid region is studied, as pictured in Figure 3. Mass flow inlet and pressure-based outlet are considered.

In the present study different elements combination were characterized, based on their mixing efficiency. For this purpose, the fluid domain was divided in two (or more for a combination of elements) cross-sectional planes, one at the beginning and one at the end of the considered elements. Both cross-sectional planes were divided into 1150 small bins. A total of 2562 massless particles were homogeneously and equidistantly distributed in the initial cross-sectional plane, one half on the top, the other half on the bottom as shown in Figure 4.

A color or index property is assigned for both streams (red/blue, or 0/1). The size of the bins and the number of the injected particles are determined in a preliminary internal study.

The particle position and flow characteristics over the time are used for the statistical evaluation of the mixing achieved by counting the number of particles over the bins over time. The concentration of each bin is calculated as follows:(4)$$c_{bin}=\frac{number\;red\;particles}{number\;red\;and\;blue\;particles}$$

Subsequently the standard deviation is built over the entire grid, as well as the variance, followed by the scale of segregation, or mixing index:(5)$$S_{c}=\sqrt{\frac{1}{m-1}\;\;\;}*\overset{m}{\underset{i=1}{\sum }}{\left(c_{i}-\bar{c}\right)}^{2}$$
(6)$$V=\frac{S_{c}}{\bar{c}}$$
(7)$$S^{2}=\frac{S_{c}}{\bar{c}}$$
where *m* is number of bins, $\bar{c}$ is mean concentration, and $c_{i}$ is local concentration for the considered bin. S is 0.5 for the fully mixed state and 1 for the fully unmixed state.

To calculate the scale of segregation over time, a special post-processing routine was developed. The commercial software ANSYS FLUENT™ is used for the numerical particle tracking. The software predicts the trajectory of a discrete phase particle (or droplet or bubble) by integrating the force balance on the particle, which is written in a Lagrange reference frame. The use of massless particles, in this case, eliminates this force and the discrete massless particle follows the flow and temperature of the continuous phase. The trajectory equations are solved by stepwise integration over discrete time steps. Integration of time yields the velocity of the particle at each point along the trajectory, with the trajectory itself predicted by [42].
(8)$$\frac{d\vec{x}}{dt}=\vec{v\;}$$

The used numerical scheme to solve the equation of motion of the particle is the trapezoidal. The massless particles are injected continuously over the time. The chosen time step for the particles is different from the time step used for the continuous phase. The particle time step size assures an equidistant position during the injection also in the fluid flow direction. 

A continuous fully filled fluid flow calculation over multiple rotations is performed. A 3D isothermal numerical simulation is carried out. The highly filled polymer (87 wt.-%) is assumed to obey the Carreau viscosity model shown, as below, where η is the dynamic viscosity and $\dot{\gamma }$ the shear rate. A, B, and C are the characteristic model parameters of the Carreau model, assumed equal to A=3 ∗ 10^5^ Pas, B = 9.1 1/s and C= 0.727, respectively.
(9)$$\eta =\frac{A}{{\left(1+B*\dot{\gamma }\right)}^{C}}$$

The viscosity data are achieved by rheological characterization of the compounds by Institut für Kunststofftechnik at the University of Stuttgart, as described in Section 2.4. 

### 2.3. Multiphase Flow

The mixture model is used for the calculation of the real mixing process due to its simplicity versus the full Eulerian model, and because of the possibility for the modeling of the granular phase for liquid–solid flows. The simulation procedure of the mixing process is described in Figure 5. The mixture model solves the continuity equation for the mixture, the momentum equation for the mixture, and the volume fraction equation for the secondary phase, as well as algebraic expression for the relative velocities (if the phases are moving at different velocities) [42].

In previous work, a three-phase model for simulation and calculating the mixing process in semi filled solid-liquid flow was developed and verified experimentally. This model is used, as follows:

Only the mixing zone is considered. The second portion (higher percentage) filler, defined as the granular phase, is added to the premixed filled polymer, which is described as a continuous phase using the Carreau viscosity model. The values for A, B, and C for a filling degree of 41 wt.-% are, respectively: A = 4.1 ∗ 10^3^ Pas, B = 0.772 1/s, and C= 0.481. The mixing process is simulated over many revolutions until reaching the stationary state. No interaction is allowed between the air phase and the other two product phases consisting of premixed polymer and powder. The mixing density is used as a physical value for confirming the stationary state and for the evaluation and comparison between the different geometries.

### 2.4. Rheological Characterization and Materials

The rheological characterization of the compounds was carried out with a plate–plate rotational rheometer (PPR), Discovery HR-2 from TA Instruments, New Castle, DE, USA, and also with a high-pressure capillary rheometer (HCR), Rheograph RG 50 from Göttfert Werkstoff-Prüfmaschinen GmbH, Buchen, Germany.

Furthermore, the melt density of all compounds is determined by the quotient of melt mass-flow rate (MFR) and the melt volume-flow rate (MVR).
(10)$$\rho =\frac{MFR}{MVR}$$

The different melt flow indices were measured by means of the melt flow testing machine MP-D from Göttfert Werkstoff-Prüfmaschinen GmbH, Buchen, Germany.

Initially, round capillaries were used to characterize the material by HCR, but this did not lead to reproducible results, and the necessary Bagley correction [43] made it difficult to evaluate the results. Therefore, two geometrically similar slit dies were used to analyze the wall slip behavior according to Geiger [25], as explained in Section 1. Three pressure transducers are installed along the flow path of the slit die. The scaling factor of the dies equals 0.83. Figure 6 shows a schematic sketch of the die, where the width is specified with b and height with h. The distance between the three pressure transducers is indicated by x and the distance of the first pressure transducer to the inlet of the die by y. The active length of the die is indicated by L.

A loading of the feed channel by hand, which has a diameter of 20 mm, did not provide reproducible results for rheological characterization despite uniform compaction of the material with a plunger. This is most likely due to a strong inhomogeneity throughout the melting process within the feed channel. This effect is usually not observed with unfilled thermoplastics. For this reason, a feed extruder was used to fill the feed channel with a homogeneous melt.

In this study, the viscosity curves were determined for a total of nine different compounds, with filler fractions containing 30, 50, 65, 70, 75, 80, 83, and 87 wt.-% graphite at 250 °C, including the raw polymer (0 wt.-%). In order to reduce the effort, the compounds with filler fractions up to 50 wt.-% were analyzed on a plate–plate rotational rheometer by oscillating shear stress. The diameter of the plate was 20 mm and the height of the shear gap was 1 mm.

The compounds with filler fractions above 50 wt.-% were analyzed on the high-pressure capillary rheometer with slit dies. In addition, the temperature effect on the viscosity was investigated for the compound with a filler fraction of 87 wt.-% for two further temperatures at 210 and 230 °C. Furthermore, two different compound recipes for the filler fraction of 87 wt.-% were investigated, which will be denoted as #1 and #2 in the following. The analyzed shear rate range is defined between 1 and 250 s^−1^. Higher shear rates are not expected during the extrusion of bipolar plates with a dimension of 150 mm × 2 mm, which can be confirmed via a simple analytical estimation. All measurements were performed three times to ensure a high reproducibility. The obtained values were subsequently arithmetically averaged.

### 2.5. Simulation Model of High-Pressure Capillary Rheometer

In accordance to the performed experiments, a 2D simulation model of the die and feed channel was developed (Figure 7) in OpenFOAM (version 5.x), an open source CFD software [44,45]. A constant velocity profile is applied at the inlet, where the profile is derived from the plunger velocity, respectively, specifying the shear rate in the HCR test. The corresponding feed zone was sufficiently extended within the simulation model so that a constant flow profile was established. The boundary conditions on the walls are chosen based on the results of the wall slip tests. In the first step of the simulations, wall adhesion of the material is assumed. According to the pressure transducer positions in the experimental setup, the simulative calculated pressure loss is evaluated along the slit die by three pressure measurement positions. Since the focus is on the pressure loss along the die length and not on the absolute pressures, a pressure of 0 bar is assumed at the outlet.

The melt flow is assumed to be isothermal, incompressible, and stationary. The flow behavior is derived from the viscosity measurements carried out at the HCR. Depending on the filling ratio, a Carreau model (acc. to Equation (9)) as well as a power law (Equation (11)) were used. In Equation (11), k is the consistency factor and n the flow exponent of the power law.
(11)$$\eta =k*{\dot{\gamma }}^{\left(n-1\right)}$$

The geometry preparation as well as the computational mesh were created by means of ANSYS ICEM CFD™. The structured mesh consisted of approximately 284,000 hexahedral elements. The simulations were performed with OpenFOAM (version 5.x) and subsequently analyzed by ParaView.

## 3. Results

In the following, the results for the compounded graphite mixtures to produce bipolar plates are presented. First, the mixing quality of the compounding process within the twin-screw extruder was evaluated on the basis of numerical simulations. Subsequently, the produced compounds were characterized rheologically by means of PPR and HCR measurements and the determined material behavior was reproduced with suitable models. Finally, the material model was applied in a flow simulation of a slit die to reproduce the HCR measurement and thus determine the existing deviation between virtual prediction and reality.

### 3.1. Simulation of Distributive Mixing Process in Co-Rotating Twin Screw Extruders

In the initial screw configuration for highly filled polymer compounding, a set of two 45° staggered wide kneading blocks were used in the mixing zone in the extruder. The target was to find a new configuration of elements, ensuring better distributive mixing between the powder and the premixed polymer. Referring to this, the mixing capability of different elements was analyzed based on the particle tracking method. These characteristics were subsequently confirmed in a multiphase semi-filled calculation.

It is well known that kneading blocks, and especially special mixing elements, are responsible for good distributive mixing in the extruder. Therefore, the mixing characteristics of those elements were investigated numerically. Table 2 gives the elements and their combinations, studied here.

To quantify the distributive mixing performance, the particle tracking method and the scale of segregation over time were applied, as described in Section 1. The number of bins of the grid and the position of the grid, as well as the number and position of the initially placed particles are identical for all studied combinations.

As a result, the evolution of the mixing over the number of revolutions was achieved, which is the same as the evolution of the mixing over time. In the following, the results for all studied combinations of elements are presented and compared.

#### 3.1.1. Mixing over Kneading Blocks

Four kneading block combinations were studied. Their geometry is shown in Figure 8. One very useful analysis for the mixing based on the particle tracking method is given by the bin concentration cut-outs. They are generated for each element after reaching the stationary state and illustrate the concentration distribution over the grid and between the bins themselves. The bin concentrations cut-outs for the studied configuration are presented in Figure 9. The comparison of the diagrams for the evolution of the mixing coefficient is shown in Figure 10, which highlights the evolution of the distributive mixing after each element of the configuration and between them. While for all configurations the use of a second element does improve the mixing, the use of a third element is mainly related to a pure redistribution between high and low concentrated bins. Quite different behavior shows the configuration of narrow disks with a staggering angle of 45°, where the third element does really improve the mixing for the studied configuration.

The direct comparison of the kneading blocks leads to the following conclusion: the best mixing characteristic offers the three-element combination of conveying (45°) narrow blocks with a value of 0.75, as well as both configurations of neutral (90°) kneading blocks; however, this includes only two of them, also reaching a value of 0.75. These configurations are an improvement on the reference one, having a value of 0.8. Again, it is important to note that the mixing value goes from 0.5 for the best mixing to 1 for the fully segregated state.

#### 3.1.2. Special Mixing Elements

Both backflow elements ZME and TME and their configurations are shown in Figure 11 and Figure 12, respectively. The ZME element produces a backflow due to the grooves in the flighted pitch profile, whereas the TME has evenly distributed left-handed grooves, resembling an impeller, and a disk as the counter element. The use of only one ZME element assures a very good mixing characteristic, comparable to those from the kneading block configurations. The next succeeding element slightly improves this behavior, reaching the best mixing characteristic value of 0.7, shown in Figure 13. On the other hand, the combination of TME elements also offers the most intensive improvement of the mixing characteristic due to the use of more elements because of its shorter primary length. The use of five elements leads to a mixing improvement of about 15%.

The calculated mixing coefficients are summarized in the Table 3 below:

The analysis of the mixing behavior for different kneading blocks and the comparison of the calculated mixing coefficients revealed some potential improved configurations, which could be used in the mixing zone during extrusion, namely two narrow neutral kneading elements or three narrow conveying kneading elements, or even two wide neutral kneading elements. They all offer a mixing improvement when compared to the reference. On the other hand, both special mixing elements TME and ZME exhibit the best mixing efficiency between the studied elements. Mixing and processing high-filled polymers is challenging not only for choosing the best mixing element or element combinations, but also to find out those producing the lowest dissipated energy that is additionally generated by the high amount of filler. This is the energy due to the inner friction of the product during the process and depends highly on its viscosity and the square of the shear rate. It is calculated as the integral value of the product of the viscosity and square of shear rate.

In Figure 14 the normalized dissipate energy for the studied elements is demonstrated. It is interesting to point out that the special mixing element TME and ZME produce the higher energy also due to the backflow effect they offer, which is excellent for the mixing characteristic but not favorable for the dissipated energy, which will additionally heat the product. For this reason, the choice of optimal mixing element for highly filled compounds is achieved as a moderate balance between a better mixing characteristic and a low dissipated energy. In this study this is fulfilled by both kneading blocks.

### 3.2. Results of the Multiphase Mixing Simulation

Two screw configurations, namely two elements of narrow 90° blocks and three elements narrow 45° kneading blocks, were chosen from the particle tracking-based mixing analysis and simulated in a multiphase calculation, as described in Section 1. These two configurations were taken because of the very good mixing efficiency they offered, combined with the lowest dissipated energy. In the multiphase simulation, the evolution of the mixing density for the considered configurations is plotted against the time. Four revolutions are calculated, corresponding to 0.4 s.

As already predicted by the particle tracking method and the evaluation of the scale of segregation, both configurations yield to similar distributive mixing, one of the best between the studied elements. This behavior was confirmed by the multiphase simulation. Both configurations reached the stationary state and thus the mixing density at about four revolutions, as shown in Figure 15.

Moreover, the slower rise of the mixing density progress for the three-element combination was in correlation with the mixing behavior achieved by the scale of segregation, namely the third element is the deciding element, which affirmed the achievement of the highest mixing characteristic. On the other hand, this is also due to the better conveying efficiency of this combination when compared to the neutral element.

This can be seen in the velocity profile distribution for the mixture for both elements shown in Figure 16. The velocity contour plot for the neutral kneading element shows clearly distinguished regions of high velocity around the tips of the elements. High velocity and more repetitive spatial displacements lead to better distributed mixing.

These two configurations were also tested in the trials and showed a similar level of quality as trials prescribed numerically. The quality evaluation consists of measuring the typical physical properties for high-filled products after compounding.

Moreover, it should be noticed that not only is the mixing efficiency to be considered, but the dissipated energy especially needs to be considered while operating with highly filled polymer. The advantageous use of conveying kneading blocks regarding their lower dissipated energy (about 20%) is recommended in this case, particularly in view of the same mixing efficiency. The high dissipated energy predicted numerically is also a reason not to use the special mixing elements for highly filled compounds, although they offer the best mixing quality (similar to TME).

### 3.3. Concluding Remarks of Mixing Simulation

Different element configurations were studied, and their mixing characteristics were evaluated using the particle tracking method. Compared to the reference configuration, some improved combinations were found. Moreover, two configurations were evaluated in a multiphase semi-filled calculation and in the experimental set-up, which confirmed the already obtained results.

In the following, the results of the rheological characterization of the produced compounds are presented.

### 3.4. Melt Density and Results of High-Pressure Capillary Measurements

The following Figure 17 shows the experimentally determined melt densities of the different filler ratios of the graphite compounds. As expected, the density increases significantly with increasing graphite content. The values vary in a range from 0.77 g/cm^3^ for the unfilled base polymer to a value of 1.75 g/cm^3^ for the highly filled compound, with a mass fraction of 87 wt.-% graphite. The density of pure graphite tends to be around 2.1 g/cm^3^. Compared to the unfilled raw material, the melt density doubles already at a filling ratio of approximately 75 wt.-%. This already suggests a processing challenge.

Figure 18 shows the determined pressure profiles of two representative compounds which have been measured at the three sensor positions along the flow path of slit die 1 by four different plunger velocities. The obtained curves show a distinct linear pressure profile, hence the values can be considered as valid measurements for the subsequent evaluations. Based on the pressure differences and the defined plunger speed, the shear velocity as well as the corresponding apparent dynamic viscosity can be calculated by means of the shear stress and the slit die geometry. To ensure the reproducibility of the results, several measurements were carried out.

The resulting apparent shear viscosities for all analyzed compounds, as well as the raw polymer, are shown in Figure 19. All measured values very clearly show the expected correlation between the degree of filling and the resulting viscosity. The higher the graphite content, the higher the resulting viscosity of the compound. Moreover, the highly filled compounds characterized by HCR all show a very linear structural viscous behavior. The observed differences are most likely related to the different measurement methods: on the one hand oscillatory shear at small deformations at PPT measurement, and on the other hand continuous shear at high pressures at HCR measurement. Furthermore, the effective ratio of wall slip effects may vary according to the measurement method. An evaluation of the wall slip rates based on these measurements is not possible. The two different compound recipes for the filler fraction of 87 wt.-% hardly show any difference in the flow behavior in the investigated shear rate range.

The influence of the temperature on the viscosity was also investigated. The results can be found in Appendix A.

### 3.5. Analysis of Wall Slip Effects

Based on the theory used to characterize wall slip effects, HCR measurements with two geometrically similar slit dies were compared. The use of the Geiger–Mooney [25] approach, the quantities determined, and a validated MATLAB based program to evaluate wall slip effects are well-described in [46].

In Figure 20 the apparent shear rate against shear stress is shown for different filler fractions of the graphite compounds. The analyzed slit die geometries according to the specifications in Figure 6 are abbreviated, denoted by sd1 and sd2. The results observed by slit die 1 are marked with rectangles, whereas the results observed by slit die 2 are marked with circles. If wall adhesion is present, the measured curves for two geometrically similar slit dies should coincide, according to the theory. In contrast to the expectations, the measured curves for high filling ratios in particular show a high coincidence. Although it is not explicitly shown in Figure 20, due to the clarity of the diagram, a high consistency can also be observed for the second formulation of the recipe of the highly filled compound with a mass fraction of 87% (#2). Since slit die 2 has the larger flow height, the measured points have to be observed at higher shear stresses than those of the smaller slit die 1. This is clearly valid for lower filler contents, while the highest graphite content with 87% mass fraction leads to a certain inconsistency. Currently, this can only be explained by measurement inaccuracies and not by the real material behavior.

In contrast, the divergence of the curves appears to increase with lower filler content. However, unlike the expectation that the wall slip ratio increases with a higher graphite content, this result indicates that the wall slip ratio of the analyzed graphite compounds is higher the lower the filler content. For the observed material behavior, a conclusive explanation based on these results is not yet possible. Therefore, further research must be conducted for this purpose.

This result in particular simplifies the design of different flow geometries for processing of the highly filled compounds into customized end products. According to the results obtained, only a very small amount of wall slip effects occur for the highly filled compounds, which may be neglected within the design process. In contrast, the effects for the low filler contents are quite large and, accordingly, we expect them to have a significant impact on the reliable prediction of the flow.

### 3.6. Simulation Results of High-Pressure Capillary Rheometer

The purpose of the following numerical investigation is to calculate the pressure loss along the slit die based on different flow approaches to determine the resulting deviation from the experimental values. Hereby, the influence of the chosen flow approach on the predicted pressure can be tested. Figure 21, Figure 22 and Figure 23 show the experimentally determined pressure losses at specific shear rates in comparison to the predicted results by means of numerical simulations. The shear rates in the simulation model are calculated according to Equations (9) and (11), whereas the apparent shear rate in the experiment results from the adjusted plunger speed of the HCR-system.

Even though the previous results have shown a high wall slip effect for the lower-filled compounds, the predicted deviation of pressure loss is quite small for all analyzed shear rates. However, the deviation in the range of high shear rates for the first recipe of the highly filled compound with 87% graphite by mass is rather conspicuous. Since the results determined by HCR measurement only provided information on the wall slip effects up to the suspicious shear rate (200 s^−1^) for this compound, the presence of wall slip effects could hardly be excluded completely. However, a plausible explanation for the abnormally high deviation for this specific simulation run has not yet been found. Nevertheless, a numerical problem could not yet be found.

## 4. Conclusions

In this study, graphite compounds for the future production of bipolar plates for fuel cells were successfully produced with a mass fraction up to 87% graphite. As expected, the melt density of the compounded materials shows a significant increase with increasing graphite content. Furthermore, the rheological investigations by means of the plate–plate rheometer and the high-pressure capillary rheometer show a significant increase in the viscosity with increasing graphite content. The curves measured with the HCR, however, show an almost linear shear thinning behavior in the entire shear rate range for all analyzed compounds compared to the PPR. The investigated temperature dependence of the viscosity does not show any remarkable peculiarities and corresponds perfectly to the expectation. In contrast, the wall slip behavior characterized according to the theory of Geiger shows a behavior deviating from the expectation. As the graphite content increases, the effect of the wall slip decreases. Only for compounds with a low graphite content is a distinct wall slip behavior observed. The highly filled compounds, on the other hand, only show slight wall slip effects. A conclusive explanation for this behavior cannot be found either in the investigations carried out here nor in existing studies in the literature.

Despite occurring wall slip effects, the developed flow simulation for the design of slit dies can provide a good prediction for both low and high filling ratios of the graphite compounds.

Different mixing element configurations were studied, and their mixing characteristics were evaluated using the particle tracking method. Compared to the reference configuration, some improved combinations were found out. Moreover, two configurations were evaluated in a multiphase semi-filled calculation, which confirmed the already obtained results performed by the particle tracking method. Based on the adopted numerical techniques, a combination of elements was chosen, offering the best mixing efficiency and the lowest dissipated energy for use in the experimental set-up. An optimized screw configuration will be used in the future and should finally confirm the results from the numerical simulation.

## Figures and Tables

**Figure 1 polymers-15-02589-f001:**
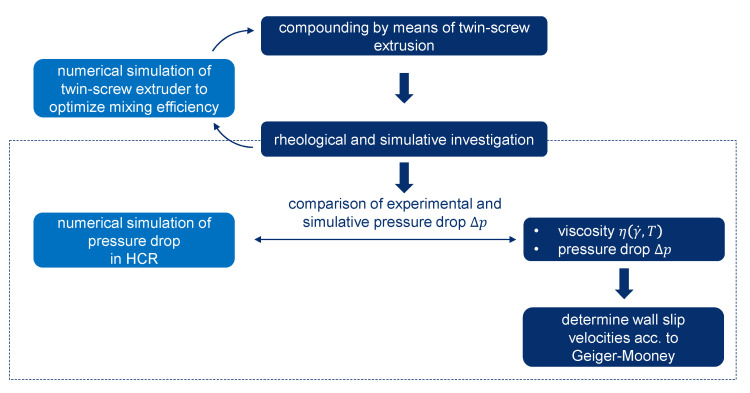
Schematic representation of the methodology within the scope of this work.

**Figure 2 polymers-15-02589-f002:**
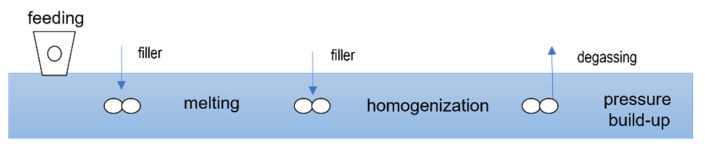
Schematic description of the experimental set-up for compounding.

**Figure 3 polymers-15-02589-f003:**
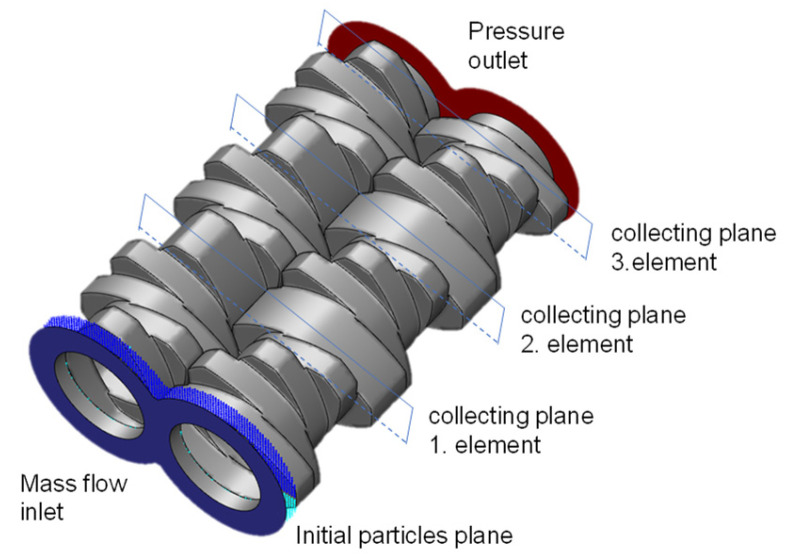
Simulation model for particle tracking.

**Figure 4 polymers-15-02589-f004:**
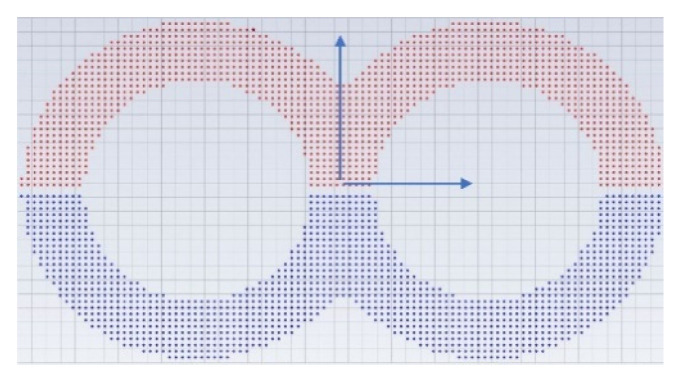
Initial distribution of the particles with grid.

**Figure 5 polymers-15-02589-f005:**
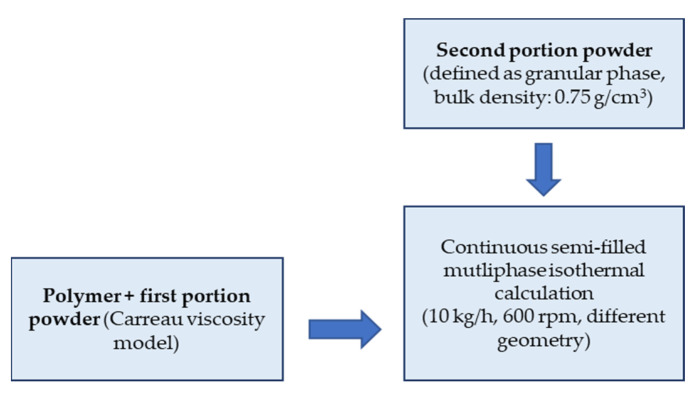
Block diagram for the multiphase simulation.

**Figure 6 polymers-15-02589-f006:**
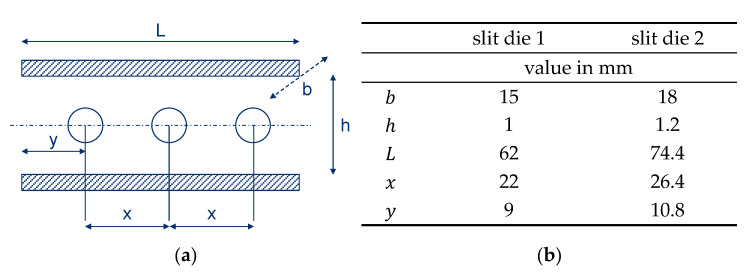
Schematic description of the geometry (**a**) and values of the geometrical parameters (**b**).

**Figure 7 polymers-15-02589-f007:**
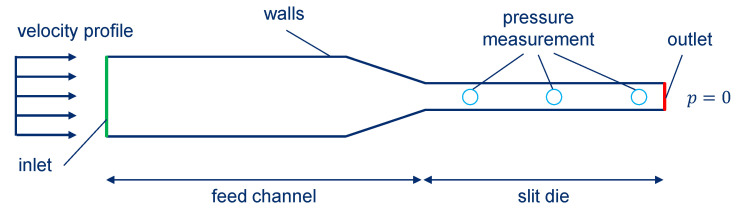
Schematic description of model used for the simulation of the high-pressure capillary rheometer.

**Figure 8 polymers-15-02589-f008:**
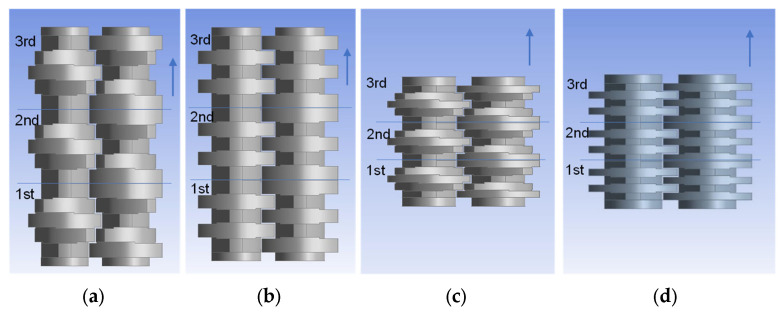
Geometry of kneading blocks, combinations of wide and narrow disks 45° and 90°. (**a**) wide disks 45°, (**b**) wide disks 90°, (**c**) narrow disks 45°, and (**d**) narrow disks 90°.

**Figure 9 polymers-15-02589-f009:**
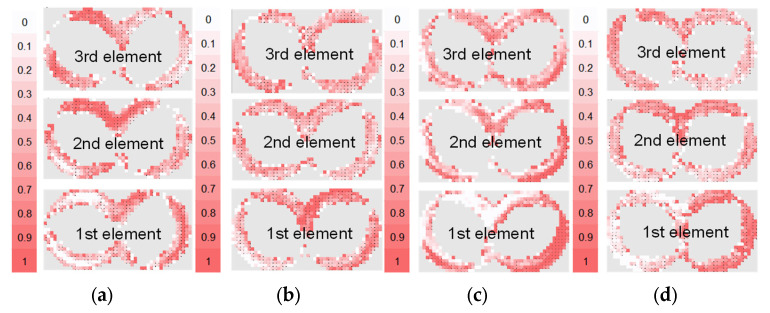
Bin concentrations distribution cut-outs (0.5: fully mixed; 1: fully segregated) after reaching the “stationary” state of mixing. (**a**) wide disks 45°, (**b**) wide disks 90°, (**c**) narrow disks 45°, and (**d**) narrow disks 90°.

**Figure 10 polymers-15-02589-f010:**
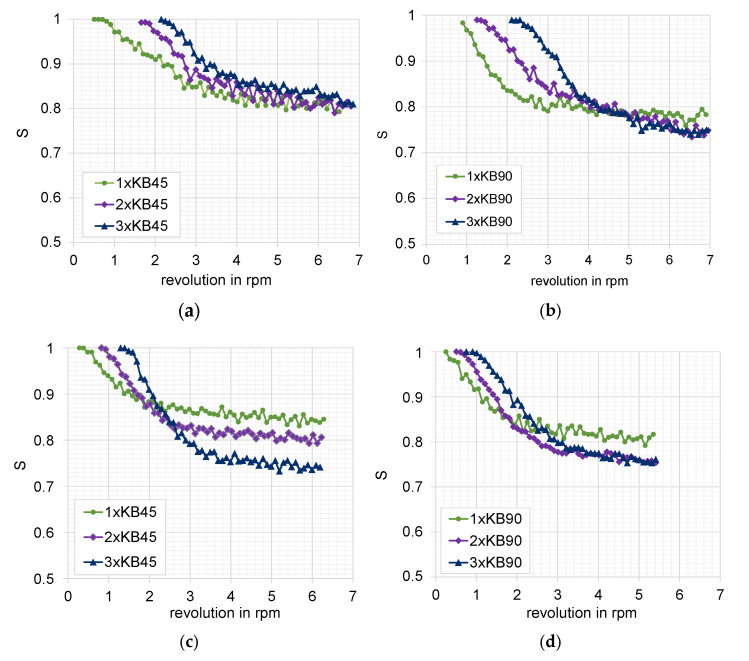
Evolution of the mixing coefficient as function of the number of revolutions: (**a**) wide disks 45°, (**b**) wide disks 90°, (**c**) narrow disks 45°, and (**d**) narrow disks 90°.

**Figure 11 polymers-15-02589-f011:**
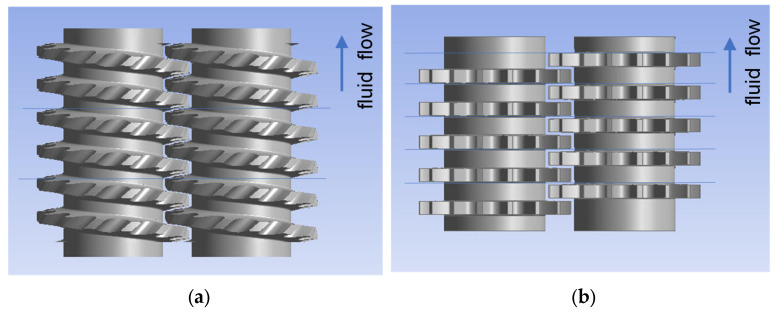
Geometry of mixing elements: (**a**) ZME and (**b**) TME.

**Figure 12 polymers-15-02589-f012:**
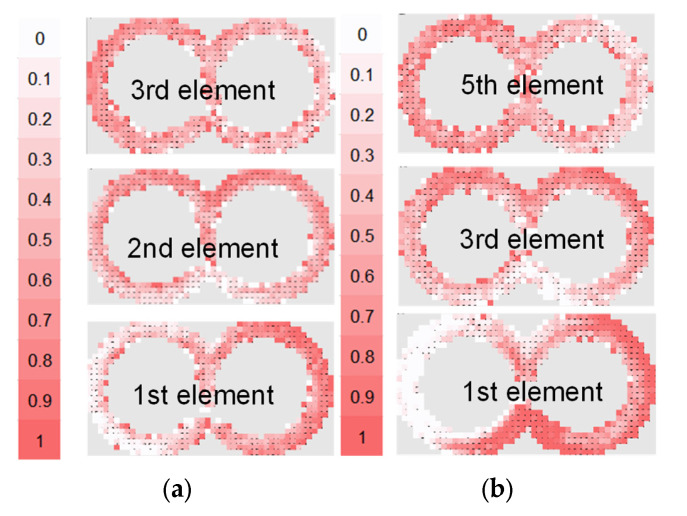
Bin concentration cut-outs: (0.5: fully mixed; 1: fully segregated) (**a**) ZME configuration and (**b**) TME configuration (only for first, third and fifth).

**Figure 13 polymers-15-02589-f013:**
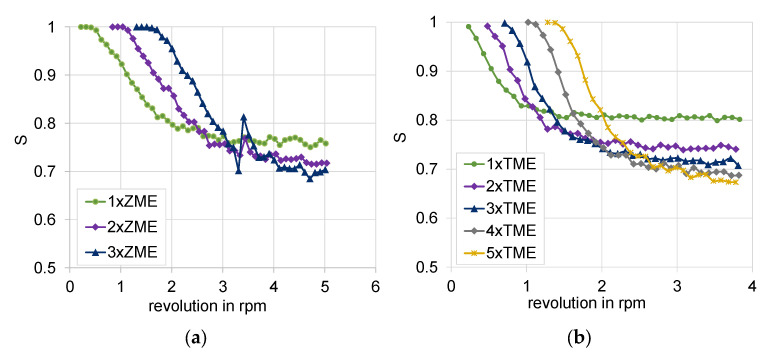
Evolution of mixing coefficient over the number of revolutions: cut-outs: (**a**) ZME configuration and (**b**) TME configuration (only for first, third and fifth).

**Figure 14 polymers-15-02589-f014:**
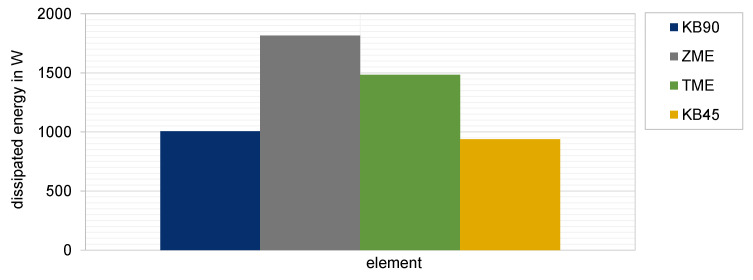
Comparison of normalized dissipated energy by different elements.

**Figure 15 polymers-15-02589-f015:**
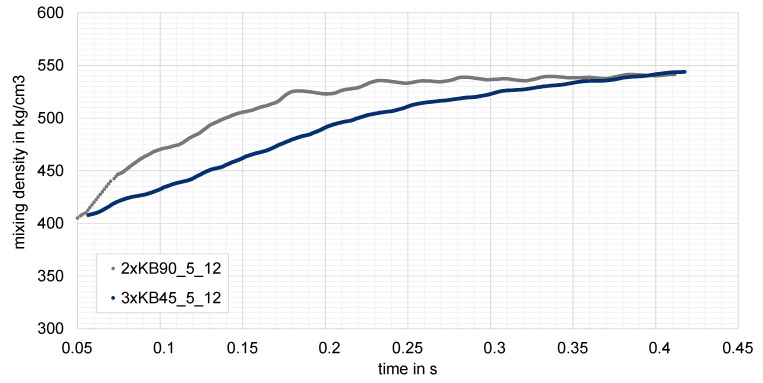
Evolution of mixing density over the time for two narrow kneading configurations: two-element neutral kneading block and three element conveying blocks. Achievement of stationary state for both at about 0.4 s.

**Figure 16 polymers-15-02589-f016:**
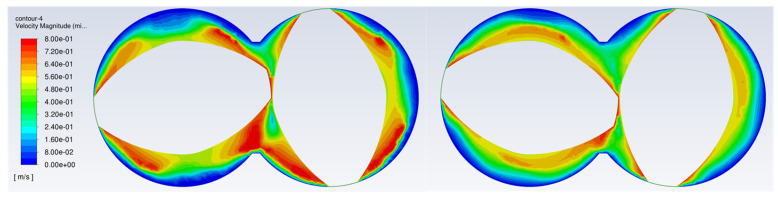
Instantaneous velocity contour plot at the end of the element combination for two-element neutral kneading blocks (**left**) and three element conveying blocks (**right**).

**Figure 17 polymers-15-02589-f017:**
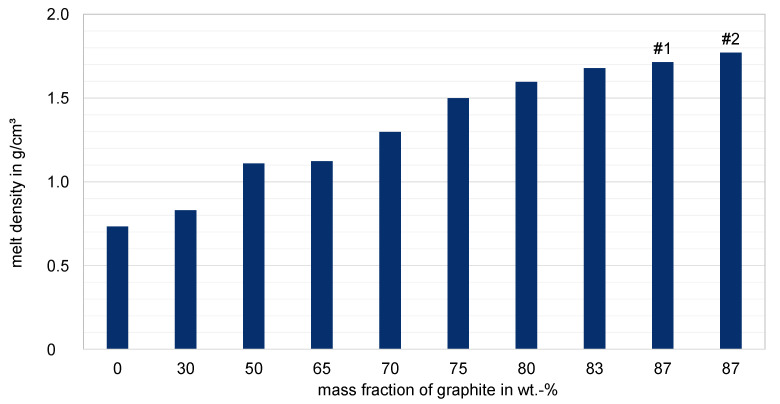
Melt densities depending on the graphite filling degree at a temperature of 250 °C.

**Figure 18 polymers-15-02589-f018:**
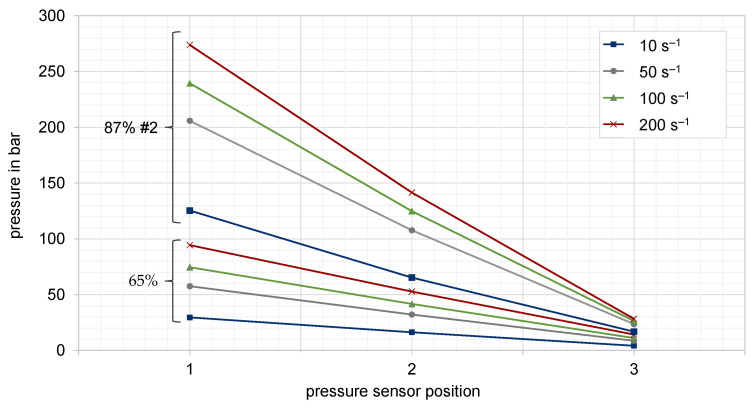
Axial pressure profiles for a graphite filling degree of 65 wt.-% and 87 wt.-%-#2 at a temperature of 250 °C (slit die 1).

**Figure 19 polymers-15-02589-f019:**
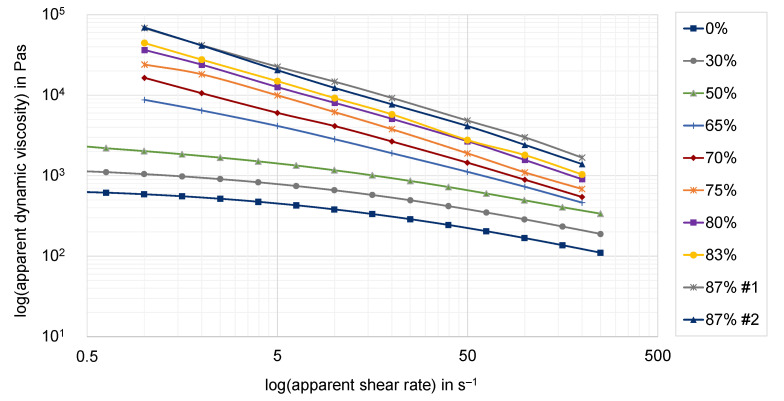
Apparent shear viscosities in dependence of the graphite mass fraction at a temperature of 250 °C (slit die 1), compounds with filler content up to and including 50 wt.-% were determined by PPR and higher than 50 wt.-% were determined by HCR.

**Figure 20 polymers-15-02589-f020:**
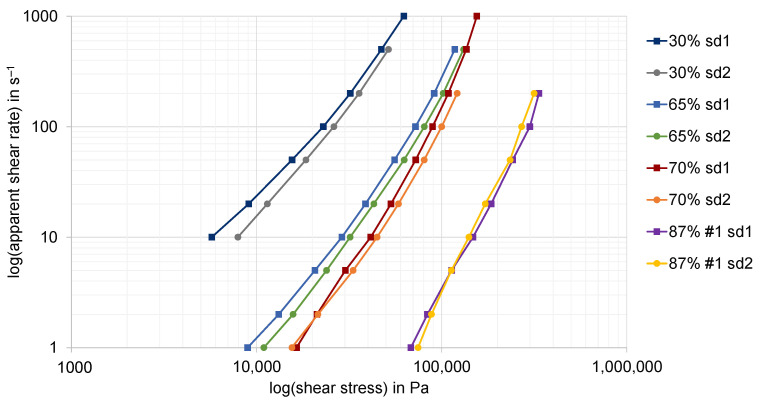
Comparison of the apparent shear stress and shear rate for two different slit dies (sd1 and sd2) in dependence of the filler fraction of the analyzed compounds at a temperature of 250 °C.

**Figure 21 polymers-15-02589-f021:**
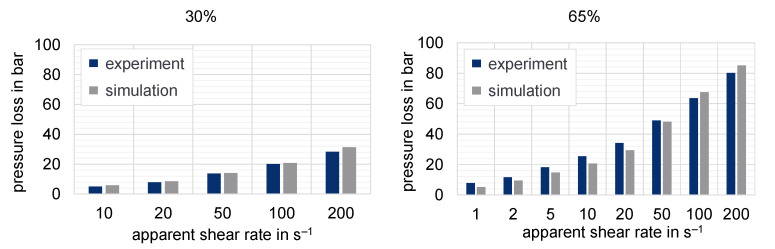
Pressure losses measured with HCR to simulation results for 30% and 65% mass fraction of graphite at a temperature of 250 °C (slit die 1).

**Figure 22 polymers-15-02589-f022:**
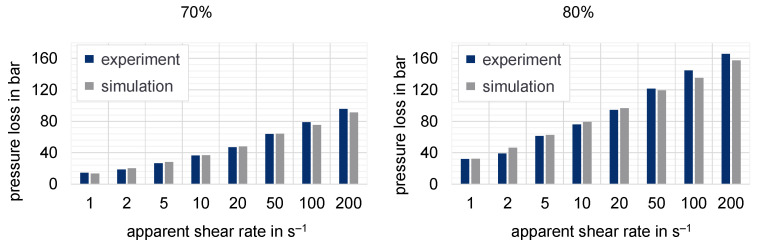
Pressure losses measured with HCR compared to simulation results for 70% and 80% mass fraction of graphite at a temperature of 250 °C (slit die 1).

**Figure 23 polymers-15-02589-f023:**
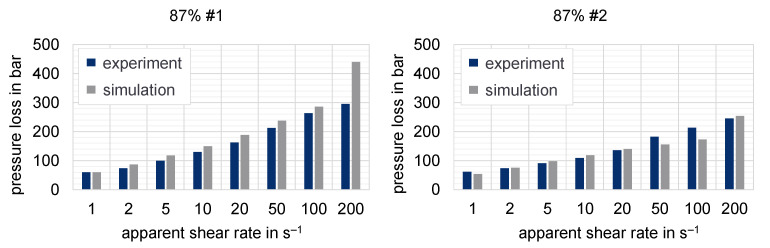
Pressure losses measured with HCR compared to simulation results for the different compounds #1 and #2 with 87% mass fraction of graphite at a temperature of 250 °C (slit die 1).

**Table 1 polymers-15-02589-t001:** Process parameters for compounding.

	Process Data
screw diameter	26 mm
throughput	10 kg/h
rotational speed	600 rpm
L/D	48

**Table 2 polymers-15-02589-t002:** Studied elements and their combinations.

Element	Staggering Angle	L/D
Zahnmischelement ZME—mixing elements	-	1 (3 elements)
Turbine Mixing Element TME—mixing element	-	1 (5 elements)
Kneading blocks—narrow disks	90°	0.5; 1; 1.5 (1–3 elements)
Kneading blocks—wide disks	90°	1; 2; 3 (1–3 elements)
Kneading blocks—narrow disks	45°	0.5; 1; 1.5 (1–3 elements)
Kneading blocks—wide disks (Reference configuration)	45°	1; 2; 3 (1–3 elements)

**Table 3 polymers-15-02589-t003:** Overview of the calculated mixing coefficient.

Element	Mixing Coefficient L/D = 1	Mixing Coefficient L/D = 1.5
ZME—mixing elements	0.7	-
TME—mixing element	0.67	-
Kneading blocks 90°—narrow disks	0.75	0.75
Kneading blocks 90°—wide disks	0.78	0.75
Kneading blocks 45°—narrow disks	0.8	0.75
Kneading blocks 45°—wide disks	0.8	0.8

## Data Availability

Not applicable.

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
