# Peer review of "Compounding, Rheology and Numerical Simulation of Highly Filled Graphite Compounds for Potential Fuel Cell Applications"

_polymers, 2023, doi:10.3390/polym15122589_

Round 1

Reviewer 1 Report (Previous Reviewer 3)

In this resubmitted version, the authors have extensively revised the work and reformulated the structures in better shape. I agree with the authors' reply that the wall slip measurement is a significant improvement in this article. Also, they quantified the simulation results by comparing with the experiments. 

However, the manuscript still seems to be a technical report rather than a typical research paper. For example, still to many figures are included. I suggest to exclude unnessary figures and move them into supplementary information. Combining some relevant figures is also a good way to polish and concise the manuscript. 

Author Response

Dear reviewer,

thank you for your comments.

Again, we revised the manuscript (see attached). In the new version, we combined to figures (former Fig. 7 and 8 to one single figure 7). Besides we shifted the temperature dependency of the viscosity to Appendix A, as it can be seen as an edge study.

Finally we combined fig. 4 a) and fig. 4b) to single figure 4.

We hope that themanuscript is now suitable for publication.

Thank you.

Kind regards

Reviewer 2 Report (Previous Reviewer 4)

The authors have improved their paper that is now suitable for publication.

English language is fine.

Author Response

Thank you very much. We are glad to hear, that the paper is now suitable for publication.

Kind regards

This manuscript is a resubmission of an earlier submission. The following is a list of the peer review reports and author responses from that submission.

Round 1

Reviewer 1 Report

This manuscript is about graphite compounds 18 with a filler mass fraction up to 87 % were successfully produced and characterized. The topic is interesting need some clarification in the form of title provided with the materials present in the paper. Methodology is not clear it seems lumps there will be consice and author should provide a block diagram which represent the whole main purpose of the study. There should be addition of some numeric data. The references are appropriate. Figures quality should be improved.

Author did good attempt need to address some observation

1. Author should add revise the abstract via the addition of numeric findings

2.  author should clarify whether this form of introduction (divided into subsection) is according to the guidline or not if not then kindly revise it as per guidelines

3. author should give block diagram to show complete metholody for current research work

4. author did almost work on simulation so there is need to add some suitable words in title so that reader could understand the difference between them

5. author need to careful proof reading

6. conclusion also revise to add some more finding 

Author Response

Dear reviewer,

thank you for your critical review and the useful advices concerning the revision of our paper. We have implemented all of them. 

Kind regards

Reviewer 2 Report

The presented manuscript of Alptekin Celik et al is devoted to the study of systems with a high content of graphite filler. At the same time, for the system under study, the behavior in a twin screw extruder and the rheological behavior using rotational rheometry are studied. The work was done at a decent level, but a little overloaded with unnecessary information and semantic repetitions. It can be seen from the literature review that the authors have familiarized themselves with the work of a number of leading rheologists and rely on their results. I don't understand what's going on with the filler. How does the morphology of the system change in the process of its deformation?! How is a good filler distribution proven? What was the distance between the plates and what was their diameter? Can the authors combine the data in Fig. 17 and 19? It is necessary to characterize the substances used in more detail.   Line 111. "1931 and 1989" recommend to delete. Lines 122-128. Why is the description of the phenomenon for systems with ceramics not suitable for systems with graphite? Lines 303-307. This statement has no scientific meaning and should be deleted, leaving only the name of the equipment used. Line 516. "results show a tendency towards a Newtonian" - not quite agree with the authors. Figure 19. I recommend swapping the axes and rebuilding the graph. Add temperature to the caption.

Author Response

Dear reviewer,

thank you for your critical review and the useful advices concerning the revision of our paper. We have implemented most of them. Please find our remarks concerning the other points below.

"I don't understand what's going on with the filler. How does the morphology of the system change in the process of its deformation?! How is a good filler distribution proven?"

  • The filler is processed in two steps during the compounding in order to be distributed homogeneously. The quality of the compound at the end of the process is proven by testing and evaluation of the compounds in respect of their physical and mechanical properties such as strength, thermal and electrical conductivity. All tested compounds revealed constant behaviour and values over the defined threshold. These results are confidential.

Can the authors combine the data in Fig. 17 and 19? 

  • The diagrams now have the figure numbers 20 and 22. Plotting the data in two different diagrams highlights better the difference between them, namely the first one showing the effect of the degree of filling on the resulting viscosity and the second one representing the wall slip effect by comparing both slit dies. A combination of both diagrams may, in our opinion, confuse the reader.

Figure 19. I recommend swapping the axes... and rebuilding the graph. Add temperature to the caption. 

  • Former Figure 19 is now Figure 22. By swapping the axes, the critical shear stress could not be read, but the critical shear stress is of huge importance for the wall slip theory. That is why we would like to keep the current form of the diagram.

We hope our comments were helpful and comprehensible.

Thank you.

Kind regards

Reviewer 3 Report

In this work, the authors have investigated a method to evaluate the mixing quality by numerical simulations of flows, and also rheological behaviors of highly filled graphite compounds to figure out wall slip behavior. 

The authors have calculated the mixing coefficient S by numerical simulations. I think the only main novelty of this work is the calculation of this effective parameters during compounding. However, I found no other novelty nor interest from this work. I suggest a rejection of this manuscript due to the following reasons.

- Abstract is too long and contains unneccessary points. 

- Introduction is too long. Please rephrase them to make more clear introduction. 

- In eq. 3, what does 'a' stand for? It is a function of shear stress at the wall, however, it is difficult to understand what the 'a' means and how can determine it.

- Overall, it seems the authors often omitted detailed information of the explanation of coefficients or parameters in the equations.

- Experimental compounding process: represent processing parameters in a table to clearly understand it. 

- In eq. 4, what is 'number i particles' and 'number all particles'? Are they 'the number of i-th particles', and 'the number of all particles'? Clarify it.

- In eq. 9, what are the eta_0, b, c, and gamma_dot? clarify them. The Carreau-Yasuda model should contain a power law index and relaxation time. One more question is, how did the authors obtain the relaxation time and power law index to apply them into the flow simulation? The authors should obtain them from the rheological experiments, however, the detail information is missing.

- Most of the references are out-dated. 

Author Response

Dear reviewer,

thank you for your critical review and the useful advices concerning the revision of our paper. We have implemented all of them. Besides, we added new references. 

We are very sorry to read that you did not find any novelty in the work, which was in our understanding the reason for rejection. Regarding the novelty of the work:

We still do believe that a novelty of the work is given and the work is of huge importance for future mobility opportunities. So far, there is no work or study dealing with wall slip characterization and modeling of such highly filled graphite compounds. We pointed this fact out in the formulation of the work's aim (l. 164-ff.). We would kindly ask you to consider this fact in your future review.

Thank you very much in advance.

Kind regards

Author Response

Dear reviewer,

thank you for your critical review and the useful advices concerning the revision of our paper. We have implemented all of them, but we have some remarks: 

to point 1.: 

As this research project is handled under the direction of our project sponsor Robert Bosch GmbH, we are not allowed to share any information about the exact composition of the materials used due to confidentiality.  However, we understand your hint very well, as wall slip effects have to be discussed in terms of the materials used.  

to point 14.: 

In the meanwhile, we added a recent publication (s. [46]), where the methodology and all needed information is well described (see l. 604-607). We would not like to reproduce the already presented methodology, as the focus of this works lays in the application and transfer of the method to new, highly-filled compounds.

Thank you very much in advance.

Kind regards